# Environmental impact and phenotypic stability in potato clones resistant to late blight *Phytophthora infestans* (Mont) de Bary, resilient to climate change in Peru

**Manuel Gastelo[1]⚬\*, Carolina Bastos[2]⚬, Rodomiro Ortiz[3]‡\*, Raúl Blas[4]‡**

**1** Postgraduate School, Universidad Nacional Agraria La Molina, Lima, Perú, **2** International Potato Center, Huancayo, Peru, **3** Swedish University of Agricultural Sciences department of Plant Breeding, Lomma, Sweden, **4** Department of Plant Science, Faculty of Agronomy, Universidad Nacional Agraria La Molina, Lima, Perú

⚬ These authors contributed equally to this work.
‡ RO and RB also contributed equally to this work.
\* mgastelo_b@hotmail.com (MG); rodomiro.ortiz@slu.se (RO)

## Abstract

Potato is one of the three most important foods in the world's diet and is staple in the Peruvian highlands. This crop is affected by late blight, a disease that if not controlled in time can decimate production. The oomycete (*Phytophthora infestans*) causing this disease is controlled using fungicides, which affect the environment and human health, another form of control is the use of resistant cultivars. 30 potato clones from the LBHTC2 population were evaluated, with the objective of selecting clones with high levels of resistance to this disease, stable for tuber yield, low environmental impact and high economic profitability. The clones were planted in three field experiments in the 2021–2022 growing season. Two experiments with and without late blight chemical control in Oxapampa and Huánuco and one experiment under normal conditions of a potato crop in El Mantaro, Junin, using randomized complete blocks with three replications. The cultivars Yungay, Amarilis and Kory were used as controls for late blight resistance and tuber yield. Late blight resistance and environmental impact were determined based on experiments with and without control in Huánuco and Oxapampa. Yield stability and economic profitability were evaluated based on information from the three experiments. Clones CIP316375.102, CIP316361.187, CIP316367.117, CIP316356.149, CIP316367.147 were the ones that presented the highest yields, high Late blight resistance, phenotypically stable for tuber yield, with low environmental impact and high economic profitability, superior to control cultivars. These clones have high potential for sustainable production systems that allow reducing environmental impact, increasing economic profitability and improving producers' living standards.

## Introduction

Potato (*Solanum tuberosum* L.) is the third most consumed food crop in the world after rice and wheat. In Peru, 330,000 ha are planted each year, contributing to farmers resilience and

**Data availability statement:** All files are available from the https://data.cipotato.org/dataset.xhtml?persistentId=doi:10.21223/BW8LWJ

**Funding:** This work, which was part of the first author's PhD thesis, was funded by the International Potato Center through USAID funding. The funders had no role in the study design, data collection and analysis, decision to publish, or preparation of the manuscript.

**Competing interests:** The authors have declared that no competing interests exist.

food security [1]. Per capita consumption in Peru has increased to 90 kg, which indicates that each year there is a greater source of income for small potato producers dedicated to family farming, a key part of sustainable rural development. Likewise, Peru has become the leading potato producer in Latin America and the Caribbean with 5.3 billion tons of potatoes per year, and a productivity of 16.5 t/ha [2], thus generating more than 110,000 jobs involving 711,313 families [3,4].

Late blight disease, caused by the oomycete *Phytophthora infestans* (Mont) de Bary, is the main threat to potato production worldwide [5,6]. When not controlled in a timely and adequate manner it can cause total loss of the crop. Due to climate change this disease can currently develop beyond 4000 meters of altitude, where it previously occurred sporadically [7]. Late blight is controlled by the frequent application of fungicides [8]. However, some strains show resistance to the fungicide metalaxyl [9]. Further, up to 20 fungicide applications are needed to control the disease in some potato production areas of Peru [10], causing a threat to the environment and human health. Another form of control is using cultivars with genetic resistance to late blight, which in addition to controlling the disease at economically profitable levels, reduce production costs, increase profitability, improving the quality of life of producers, and contribute to reducing the environmental impact and human health through reduced use of fungicides [11–14]. Integrated disease management is a control method that combines these two methods with cultural tasks such as planting density, fertilization, use of quality seed, and crop rotation.

Resistance to late blight can be vertical or specific to some races of the pathogen, and horizontal field resistance or not specific to any race. To select potato clones with horizontal resistance to late blight, it is necessary to conduct field trials in several environments to account for the variability of the pathogen and the existence of various isolates in endemic areas, such as Huánuco, Oxapampa that have adequate conditions to evaluate the incidence of late blight [15–18].

The use of fungicides in higher doses to control the late blight causes the pathogen to acquire resistance and spread rapidly in the crop plot and its neighbors, recommending replacing chemical control as the main control method due to the risk it represents for public health and its impact on the environment [19]. The environmental impact in the integrated control was found to be 2% of the conventional method in The Netherlands [20]. This difference was due to the choice of the agent, volume used and reduction of drift. In Ecuador, the susceptible cultivars Capiro and Superchola had the highest rates of environmental impact compared to the clones CIP387205.5 and CIP386209.10, whose resistant to late blight reduced the environmental impact rate by 92.22% [21]. The high environmental impact rate in these cultivars is attributed to the high use of fungicides to control the disease and that this can be reduced by using resistant cultivars that reduce the number of applications and the use of less toxic fungicides. The methodology to determine the Environmental Impact Rate (EIR) was developed by Cornell University and is an indicator to evaluate the potential risk of pesticide use in resistant clones compared to susceptible cultivars [22–24]. It is very important that genotypes with resistance to late blight are stable in tuber yield across various environments [25,26]. The yield and quality of tubers is influenced by genetic, environmental, agronomic management factors and the interrelationship between them. Several investigations show that the identification of superior genotypes is complicated by the genotype × environment (GE) interaction [27]. The analysis of GE interaction and the estimation of phenotypic stability has been studied in many crops [28]. There are many statistical methods to study the GE interaction and phenotypic stability in potato [29,30]. The multivariate method additive effects and multiplicative interaction (AMMI) combines the analysis of variance with the GE interaction effects and presents the results in biplots that allow a better identification of the most

phenotypically stable genotypes [31–33]. Research carried out in Cuba and Uganda using the AMMI model demonstrated that this method is very useful in potato cultivation to identify the most stable genotypes [34,35].

The objective of this study was to determine the environmental impact and phenotypic stability in potato clones with resistance to late blight and to select clones with high levels of resistance to late blight, low environmental impact, phenotypically stable for tuber yield and high economic profitability.

## Materials and methods

Thirty potato breeding clones belonging to the LBHTC2, developed by the genetic improvement program of the International Potato Center (CIP) were evaluated. These elite clones were selected for their resistance to late blight from 2017 to 2021, under natural disease conditions in Oxapampa at 1850 masl, where the environmental conditions of temperature (15–19ºC), relative humidity > 80% and precipitation ( > 1000 mm per year), are optimal to induce a high disease pressure, in addition to the presence of complex isolates of *P. infestans*, Resistance was determined based on low AUDPC and sAUDPC values. Three cultivars were used as controls, namely Amarilis (moderately resistant), Yungay (susceptible) and Kory (resistant) (Table 1).

The trials were planted during the 2021–2022 growing season, under natural field conditions. In Oxapampa and Huánuco two trials were planted, one with late blight control (Experiment 1) and another without control (Experiment 2), which are two sites with optimal environmental conditions (precipitation, temperature and relative humidity) for a high disease pressure, in addition to a great variability of the pathogen [16,36]. A third trial was carried out in El Mantaro (Junín) under normal conditions of a marketable potato production field (Experiment 3), (Table 2). The two experiments with and without late blight control were planted in Huánuco on 22nd September 2021, and harvested on 10th February 2022, and in Oxapampa the planting was on 1st October 2021, and the harvest was on 1st February 2022.

**Table 1. Clones of the LBHT population cycle 2 with resistance to Late blight used in this study 2021–2022.**

| # | Clone | Female parent | Male parent | # | Clone | Female parent | Male parent |
|---|---|---|---|---|---|---|---|
| 1 | CIP316344.165 | CIP398098.570 | CIP398208.219 | 18 | CIP316361.191 | CIP398201.510 | CIP398208.620 |
| 2 | CIP316346.204 | CIP398192.553 | CIP398208.219 | 19 | CIP316361.209 | CIP398201.510 | CIP398208.620 |
| 3 | CIP316352.122 | CIP398098.203 | CIP398208.219 | 20 | CIP316361.244 | CIP398201.510 | CIP398208.620 |
| 4 | CIP316352.152 | CIP398098.203 | CIP398208.219 | 21 | CIP316365.166 | CIP304081.44 | CIP398208.620 |
| 5 | CIP316353.148 | CIP398190.200 | CIP398208.219 | 22 | CIP316367.117 | CIP398190.200 | CIP398208.620 |
| 6 | CIP316353.741 | CIP398190.200 | CIP398208.219 | 23 | CIP316367.118 | CIP398190.200 | CIP398208.620 |
| 7 | CIP316354.112 | CIP398208.505 | CIP398208.219 | 24 | CIP316367.134 | CIP398190.200 | CIP398208.620 |
| 8 | CIP316354.169 | CIP398208.505 | CIP398208.219 | 25 | CIP316367.147 | CIP398190.200 | CIP398208.620 |
| 9 | CIP316355.162 | CIP398208.670 | CIP398208.219 | 26 | CIP316367.148 | CIP398190.200 | CIP398208.620 |
| 10 | CIP316356.149 | CIP302551.26 | CIP398208.219 | 27 | CIP316367.177 | CIP398190.200 | CIP398208.620 |
| 11 | CIP316358.214 | CIP398098.65 | CIP398208.620 | 28 | CIP316375.101 | CIP398201.510 | CIP398203.5 |
| 12 | CIP316360.241 | CIP398192.553 | CIP398208.620 | 29 | CIP316375.102 | CIP398201.510 | CIP398203.5 |
| 13 | CIP316361.118 | CIP398201.510 | CIP398208.620 | 30 | CIP316387.156 | CIP398192.553 | CIP398208.33 |
| 14 | CIP316361.121 | CIP398201.510 | CIP398208.620 | 31 | Amarilis | | |
| 15 | CIP316361.158 | CIP398201.510 | CIP398208.620 | 32 | Kory | | |
| 16 | CIP316361.187 | CIP398201.510 | CIP398208.620 | 33 | Yungay | | |
| 17 | CIP316361.190 | CIP398201.510 | CIP398208.620 | | | | |

**Table 2. Sites where the trials were carried out. 2021–2022 growing season.**

| Site | Experiment | Altitude masl | Latitude | Longitude | Temperature average ºC | Relative humidity % | Rainfall mm |
|------|-----------|---------------|----------|-----------|------------------------|---------------------|-------------|
| Oxapampa | 1,2 | 1850 | 10°34′48″ S | 75°24′0″ W | 18.54 | 88.74 | 660 |
| Huanuco | 1,2 | 2110 | 9°48′5.9″ S | 76°4′13.26″ O | 14.76 | 84.42 | 433 |
| El Mantaro | 3 | 3320 | 11°49′20″S | 75°23′31″O* | 11.00 | 71.00 | 316 |

* Temperature, relative humidity and precipitation of the meteorological station installed during the period of field tests Source: CLIMATE-DATA.ORG (http://es.climate-data.org/location/4353/).

MINEM (http://www.minem.gob.pe/minem/archivos/file/DGGAE/ARCHIVOS/estudios/EIAS%20-%20hidrocarburos/108/EIA%20DIGITAL/Cap%203A%20LB%20Fisicoquimico/Cap%203A%20Texto.p).

The randomized complete block design was used with three replications of 10 plants per clone. The synthetic fertilizer dose of 180–200–160 kg of NPK per hectare was used, being urea (46% N) the source of Nitrogen, diammonium phosphate (46% $P_2O_5$, 18% N) the phosphorus source, and potassium sulfate (50% $K_2O$) the source of potassium. Pest control was that of a marketable potato crop.

In experiment 1, a contact fungicide (Mancozeb) was applied twice up to 35 days after planting to the clones and control varieties. In experiment 2 with late blight control, contact and systemic fungicides (Mancozeb, Cymoxanil and Propineb) were applied to the clones and varieties in an appropriate and timely manner according to the present environmental conditions and their resistance. The experiment 3 carried out in El Mantaro was planted on 27th November 2021 and harvested on 20th April 2022, No fungicide applications were carried out. In all experiments at harvest, the number of plants, as well as the number and weight of marketable and non-marketable tubers in kg per experimental unit were recorded. With these values, the marketable and total yield per hectare.

## Selection of clones resistant to late blight

30 elite clones previously evaluated for late blight resistance from 2017–2021 were evaluated. Late blight resistance was determined and validated based on the evaluations of the damage caused by late blight in the uncontrolled experiments, planted in Huánuco and Oxapampa, the AUDPC [37] and sAUDPC were calculated as parameters of the resistance of the clones. The additive linear model used for the analysis of variance was the following:

$$Y_{ij} = \mu + \alpha_i + k_j + \varepsilon_{ij}$$

where $Y_{ij}$ is the value in the plot corresponding to the $i^{th}$ genotype in the $j^{th}$ block, $\mu$ is the general mean, $\alpha_i$ is the effect of the $i^{th}$ genotype, $k_j$ is the effect of $j^{th}$ block, and $\varepsilon_{ij}$ is the experimental error (pure and residual) associated with observation $Y_{ij}$.

The AUDPC was estimated as:

$$AUDPC = \sum_{i=1}^{n} \frac{X_{i+1} + X_i}{2} \times \left(T_{i+1} - T_i\right)$$

where Xi is the percentage of infection at i days after planting, Xi + 1 is the percentage of infection at i + 1 days after planting, and (Ti + 1 − Ti) is the number of days between late blight evaluations and n is the number of evaluations.

The late blight susceptibility scale (sAUDPC) has values from 0 to 9, with 0 being a very resistant genotype and 9 being very susceptible [38], and is calculated using the following equation:

$$Sx = Sy(Dx/Dy)$$

where Sx is the scale value calculated for the clone under study; Sy is the scale value (6) assigned to the susceptible control cultivar Yungay, Dx is the AUDPC value of the clone under study, and Dy is the AUDPC value of the susceptible control.

The analysis of variance and Tukey's mean comparison test at 5% ($P = 0.05$) were performed for AUDPC, sAUDPC, marketable yield and total tubers per hectare. The statistical software SPSS version 25, R version 4.3.2 and Microsoft Excel 97-2003 were used for statistical analysis.

## Environmental impact rate (EIR)

The information obtained in experiment 2 planted in Huánuco and Oxapampa was used to determine the environmental impact rate (EIR). The name of the fungicide used was recorded with its environmental impact coefficient (EIC), number of applications (NA), dose (D) and percentage of active ingredient (PAI) [22,24]. Environmental impact rate (EIR) was determined using the following formula [22]:

$$EIR = EIC \times PAI \times D \times NA$$

The environmental impact coefficient (EIC) was determined using the following formula:

$$EIC = \left\{ C\left[(DT \times 5) + (DT \times P)\right] + \left[\left(C \times ((S+P)/2) \times SY\right) + (L)\right] \right.$$
$$\left. + \left[(F \times R) + \left(D \times ((S+P)/2) \times 3\right) + (Z \times P \times 3) + (B \times P \times 5)\right] \right\}/3$$

where C is chronic toxicity, DT is dermal toxicity, P is half-life on plant surface, S is residue half-life in soil, SY is systematicity, L is leaching potential, F is fish toxicity, R is surface loss potential, D is bird toxicity, Z is bee toxicity, and B is toxicity to beneficial arthropods.

## Economic profitability

Economic Profitability was determined based on yield, production costs and farm sale price for each location and then calculated from the average value and sensitivity analysis of costs and yields using the following formulas:

$$\text{Total income} = \text{Tuber yield per hectare x farm sale price.}$$

$$\text{Net income} = \text{Total income} - \text{production costs}$$

$$\text{Economic profitability\%} = \text{Net income} / \text{production costs}100$$

The farm sale price is S/1.00 for the Yungay, Amarilis and breeding clones, for the Kory, el precio de venta fue S/ 0.70 which has a lower price due to its high glycoalkaloid content induced low quality.

The selection of the resistant elite clones was based on their AUDPC and sAUDPC, which should have been at least lower than the resistant Kory cultivar control, the marketable and total yield per hectare on average in the two locations, higher than Kory, with a low environmental impact rate and high economic profitability.

## Phenotypic stability of the marketable yield of tubers

The phenotypic stability of 30 potato clones and three control cultivars planted In experiments 2 and 3, was determined using the AMMI model [39,40], which integrates the analysis of variance, analysis of the principal components (PCs), the SVAMMI stability value AMMI [41]and the marketable yield stability index (MYSI) [42]. The model used for the analysis of variance was the following:

$$Y_{ij} = \mu + g_i + e_j + \sum \lambda_k \alpha_{ik} \gamma_{jk} + \epsilon_{ij}$$

where, $Y_{ij}$ is the marketable performance of the $i^{th}$ clone in the $j^{th}$ environment, $g_i$ is the effect of the $i^{th}$ clone, $e_j$ is the effect of the $j^{th}$ environment, $\lambda_k =$ is the square root of the value of the $k^{th}$ principal component, $\alpha_{ik}$ and $\gamma_{jk}$ are the $k^{th}$ principal component of the $i^{th}$ clone in the $j^{th}$ environment, respectively, and $\varepsilon_{ij} =$ is the experimental error (pure and residual).

The SVAMMI value was used as a quantitative measure of clone stability for marketable tuber yield using the formula proposed by [41]; a clone is considered stable when its SVAMMI value is low.

$$SVAMMI = \left( \left( \text{sum of squares PC1} / \text{sum of squares PC2} \right) \left( \text{value PC1} \right) 2 + \left( \text{value PC2} \right) 2 \right) \tfrac{1}{2}$$

To select a stable clone with high marketable tuber yield, the Marketable Yield Stability Index (MYSI) was used. The lower this value indicates that the clone is stable with high marketable tuber yield. The **MYSI** was calculated as follows [42]:

$$MYSI = RSVAMMI + RMY$$

where RSVAMMI is the Ranking of SVAMMI and RMY is the ranking of marketable yield. A clone was considered stable when its MYSI was lower than that of the Yungay cultivar.

## Results

### Late blight resistant breeding clones

In the experiment 1, there were statistically significant differences ($\alpha = 0.01$) between the clones for AUDPC, sAUDPC, marketable yield and total yield, thereby indicating that the clones presented different levels of resistance and tuber yield both in Huánuco and in Oxapampa. The coefficients of variability were low (Table 3).

In Huanuco twenty-three of the 30 breeding clones had lower AUDPC and sAUDPC values than the resistant Kory with 150 and 0.53 of AUDPC and sAUDPC respectively. In Oxapampa, the AUDPC of the clones ranged from 111 to 519, all with values lower than the resistant Kory (AUDPC = 556). The sAUDPC values of the clones ranged from 0.39 to 1.84, which were lower than that of Kory (1.97). The susceptible cultivar Yungay had the highest AUDPC in Huanuco and Oxapampa with 1248 and 1633 respectively, in both sites, the susceptible Yungay had a sAUDPC of 6 (Table 4, Fig 1).

The marketable tuber yield in Huánuco ranged from 18.70 to 65.86 t/ha. Twenty-three clones had higher yield than the resistant Kory (37.65 t/ha). The susceptible Yungay had a yield of 9.01 t/ha. In Oxapampa, the clones had a marketable tuber yield ranging from 13.98 to 59.86 t/ha. Twenty-nine of them were superior in yield than Kory (17.05 t/ha). The susceptible Yungay yielded 1.02 t/ha (Table 4). On average, the marketable yield of the breeding clones

**Table 3. Analysis of variance by location for area under disease progress curve (AUDPC), late blight susceptibility scale (sAUDPC), marketable and total tuber yield (t/ha), without and with late blight control (Huánuco and Oxapampa, 2021–2022).**

| Source of variation | DF[1] | Mean Square Without Control | | | | | | | |
|---|---|---|---|---|---|---|---|---|---|
| | | Huánuco | | | | Oxapampa | | | |
| | | AUDPC | sAUDPC | Marketable tuber yield | Total tuber yield | AUDPC | sAUDPC | Marketable tuber yield | Total tuber yield |
| Repetitions | 2 | 1588.13 | 0.02 | 52.78 | 67.61 | 4457.10 | 0.06 | 146.09** | 150.56** |
| Clones | 32 | 254114.84** | 3.195** | 643.52** | 816.45** | 292319.97** | 3.68** | 545.62** | 569.63** |
| Error | 64 | 1209.23 | 0.02 | 59.91 | 47.74 | 2201.96 | 0.03 | 23.55 | 25.78 |
| C.V. % | | 20.65 | 20.37 | 17.75 | 13.47 | 11.10 | 11.15 | 13.56 | 13.03 |
| Source of variation | DF[1] | Mean Square With control | | | | | | | |
| | | Huánuco | | | | Oxapampa | | | |
| | | AUDPC | sAUDPC | Marketable tuber yield | Total tuber yield | AUDPC | sAUDPC | Marketable tuber yield | Total tuber yield |
| Repetitions | 2 | 0.01 | 0.12 | 402.92** | 804.88** | 6093.91** | 30.08** | 11.44 | 27.99 |
| Clones | 32 | 186.90** | 4.17** | 391.19* | 482.45* | 821.1** | 40.61** | 254.6** | 233.32** |
| Error | 64 | 15.96 | 0.08 | 76.24 | 92.38 | 804.98 | 3.46 | 34.59 | 39.9 |
| C.V. (%) | | 28.6 | 25.7 | 17.43 | 15.99 | 18.9 | 17.58 | 15.66 | 4.69 |

[1] DF: degrees of freedom.

** indicates significant source of variation for clones at $P \leq 0.01$.

* indicates significant source of variation for clones at $P \leq 0.05$.

was 45.68 t/ha Huánuco, higher than the yield in Oxapampa (38.47 t/ha), which could be because Huánuco had less late blight infection than Oxapampa, as indicated by the AUDPC values (Table 6). Resistant clones tend to become more infected as environmental conditions become favorable for late blight, but without reaching AUDPC values that affect yields economically. Susceptible clones on the other hand, when exposed to high disease pressure, can reach 100% infection and therefore total yield loss.

In the experiment 2, there were statistically significant differences (α = 0.01) between the clones for AUDPC, sAUDPC in both sites Huanuco and Oxapampa, statistically significant differences were found in Huánuco (α = 0.05) and in Oxapampa (α = 0.01) for marketable and total tuber yield (Table 3).

Marketable tuber yield in Huánuco ranged from 25.12 to 67.04 t/ha. Twenty breeding clones had higher yields than Kory, Amarilis and Yungay. In Oxapampa, the marketable yield ranged from 20.06 to 57.41 t/ha. Twenty-six clones had higher yields than the control cultivars, whose yields were high because late blight was controlled in a timely and adequate manner. The increase in yields in most clones was not significant when the disease was controlled. On average, in Huánuco and Oxapampa, marketable yields increased by 16.29 and 1.72% respectively, thereby showing the effect of their different levels of resistance. In the susceptible Yungay the increase was significant in Huánuco and Oxapampa; i.e., 404.11% (or 9.01 to 45.43 t/ha), and 2701.18 (or 1.02 to 28.70 t/ha), respectively. In the moderately resistant Amarilis, the increase was also significant but in a lower percentage than Yungay. In Huánuco, the marketable yield increased by 242.79% (or from 14.14 to 48.46 t/ha) and in Oxapampa it was 371.60% (or 6.09 to 28.70 t/ha). There is a high correlation between AUDPC values and marketable tuber yield increases (r = 0.75, Pearson correlation p <= 0.01) (Table 5),

## Environmental impact rate (EIR)

Late blight control in clones was carried out with Mancozeb, a contact fungicide, applied 2 to 4 times according to the presence of disease symptoms in the clones. In control

**Table 4. Tukey mean comparison test (α = 0.05) for area under disease progress curve (AUDPC), late blight susceptibility scale (sAUDPC), without late blight control (Huánuco and Oxapampa 2021–2022).**

| Clone | Huánuco | | | | Oxapampa | | | |
|---|---|---|---|---|---|---|---|---|
| | AUDPC | α = 0.05 | sAUDPC | α = 0.05 | AUDPC | α = 0.05 | sAUDPC | α = 0.05 |
| CIP316344.165 | 0 | a | 0.00 | a | 450 | hij | 1.59 | hij |
| CIP316346.204 | 0 | a | 0.00 | a | 222 | bc | 0.79 | abc |
| CIP316352.122 | 637 | e | 2.26 | e | 403 | defghij | 1.43 | defghij |
| CIP316352.152 | 0 | a | 0.00 | a | 378 | defghi | 1.34 | defghi |
| CIP316353.148 | 35 | ab | 0.12 | ab | 379 | defghi | 1.35 | defghi |
| CIP316353.741 | 23 | ab | 0.08 | ab | 274 | bcde | 0.97 | bcde |
| CIP316354.112 | 0 | a | 0.00 | a | 438 | fghij | 1.55 | fghij |
| CIP316354.169 | 0 | a | 0.00 | a | 426 | efghij | 1.51 | efghij |
| CIP316355.162 | 0 | a | 0.00 | a | 452 | hij | 1.60 | hij |
| CIP316356.149 | 150 | c | 0.53 | c | 374 | cdefghi | 1.33 | cdefghi |
| CIP316358.214 | 35 | ab | 0.12 | ab | 444 | ghij | 1.57 | ghij |
| CIP316360.241 | 0 | a | 0.00 | a | 251 | bcd | 0.89 | abcd |
| CIP316361.118 | 35 | ab | 0.12 | ab | 289 | bcdef | 1.02 | bcdef |
| CIP316361.121 | 390 | d | 1.38 | d | 376 | defghi | 1.33 | defghi |
| CIP316361.158 | 492 | d | 1.74 | d | 443 | fghij | 1.57 | fghij |
| CIP316361.187 | 12 | ab | 0.04 | ab | 399 | defghi | 1.42 | defghi |
| CIP316361.190 | 0 | a | 0.00 | a | 415 | efghij | 1.47 | efghij |
| CIP316361.191 | 0 | a | 0.00 | a | 175 | ab | 0.62 | ab |
| CIP316361.209 | 493 | d | 1.75 | d | 362 | cdefgh | 1.28 | cdefgh |
| CIP316361.244 | 17 | ab | 0.06 | ab | 111 | a | 0.39 | a |
| CIP316365.166 | 472 | d | 1.67 | d | 368 | cdefghi | 1.30 | cdefghi |
| CIP316367.117 | 12 | ab | 0.04 | ab | 274 | bcde | 0.97 | bcde |
| CIP316367.118 | 0 | a | 0.00 | a | 356 | cdefgh | 0.70 | ab |
| CIP316367.134 | 0 | a | 0.00 | a | 199 | ab | 1.03 | bcdefg |
| CIP316367.147 | 35 | ab | 0.12 | ab | 291 | bcdefg | 1.14 | bcdefgh |
| CIP316367.148 | 0 | a | 0.00 | a | 321 | bcdefgh | 1.26 | cdefgh |
| CIP316367.177 | 445 | d | 1.58 | d | 298 | bcdefgh | 1.06 | bcdefgh |
| CIP316375.101 | 12 | ab | 0.04 | ab | 175 | ab | 0.62 | ab |
| CIP316375.102 | 12 | ab | 0.04 | ab | 397 | defghi | 1.41 | defghi |
| CIP316387.156 | 117 | bc | 0.41 | bc | 519 | ij | 1.84 | ij |
| Kory | 150 | c | 0.53 | c | 556 | j | 1.97 | j |
| Amarilis | 752 | f | 2.67 | f | 1499 | k | 5.32 | k |
| Yungay | 1248 | g | 6.00 | g | 1633 | k | 6.00 | k |
| Standard deviation | 20.08 | | 0.08 | | 27.09 | | 0.10 | |

Different letter after the mean value indicates a significant difference according to Tukey test (P<0.05).

varieties, late blight was controlled with applications of Mancozeb 2 to 5 times and a systemic fungicide based on cymoxanil and propineb 6 to 7 times during the vegetative period (Table 6).

The EIR of late blight resistant clones in Huánuco ranged from 46.72 to 70.08, while it was 289.73 in the susceptible Yungay and 145.54 in the moderately resistant Amarilis. In Oxapampa the breeding clones had an EIR in the range of 46.72 to 93.74, which was lower than that of Yungay and Amarilis with 193.60 and 218.30, respectively (Table 6).

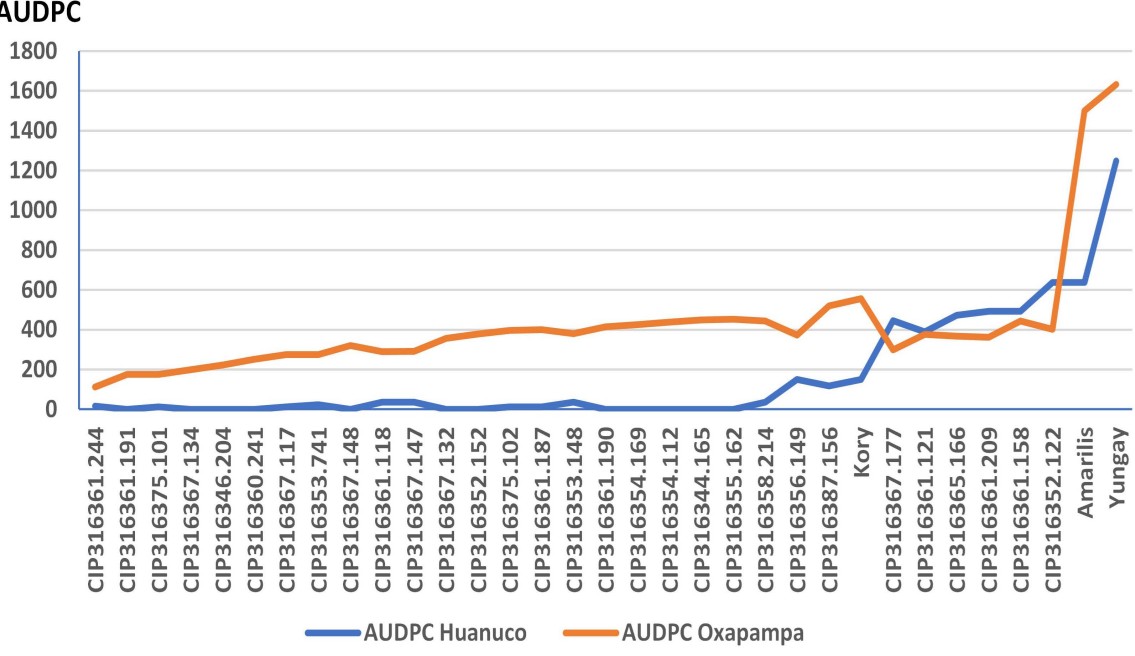

**Fig 1. Resistance to late blight (AUDPC) in potato clones.** Huánuco and Oxapampa 2021–2022.

Compared with the EIR of the susceptible variety Yungay, the clones with resistance to late blight reduced their EIR from 68 to 82%, the control variety Kory presented an EIR of 72% and the Amarillis variety 33%.

## Economic profitability

The average economic profitability of the clones resistant to late blight was 189.08% and for Kory, Amarilis and Yungay it was 70.46, 45.45 and 36.50%, respectively. The profitability of the breeding clones was therefore 2.5 to 3 times higher than the controls (Table 7). The lower use of fungicides, less labor and high yields influenced the late blight resistant breeding clones to have a high economic profitability. The sensitivity analysis of costs and yields shows us that the clones must have a minimum yield of 1701.98 kg/ha, while the control cultivars must yield a minimum of 15212.58 kg/ha to prevent loss.

## Phenotypic stability of marketable tuber yield

The analysis of variance of the AMMI model for marketable tuber yield (Table 8) shows statistically significant differences ($P = 0.01$) for the sources of variation environments, clones and their interaction (GE), thus indicating that at least one clone has different behavior in the three sites where they were evaluated. This source of variation is very important to determine through the AMMI analysis how many clones have a low GE interaction and are more stable than the others. The principal components PC1 and PC2 were statistically different in the AMMI analysis and explain 61.9% and 38.1%, respectively, of the total GE interaction (Fig 2).

The clones are considered stable when the values of the principal components are lower and tend to approach zero. The information obtained through the principal components PC1 and PC2 do not provide a quantitative measure to classify the clones by their phenotypic stability for the yield of marketable tubers. For this reason, the SVAMMI and the MYSI were used to determine which clones are stable and have the highest yields than the susceptible

**Table 5. Marketable and total tuber yield, without and with late blight control in Huánuco and Oxapampa (2021–2022).**

| Clone | Marketable tuber yield t/ha — Huanuco | | | | | Marketable — Oxapampa | | | | | Total tuber yield t/ha — Huanuco | | | | | Total — Oxapampa | | | | |
|---|---|---|---|---|---|---|---|---|---|---|---|---|---|---|---|---|---|---|---|---|
| | Without control | α=0.05 | With control | α=0.05 | Increase % | Without control | α=0.05 | With control | α=0.05 | Increase % | Without control | α=0.05 | With control | α=0.05 | Increase % | Without control | α=0.05 | With control | α=0.05 | Increase % |
| CIP316367.117 | 57.35 | abcde | 58.09 | abcd | 1.29 | 59.86 | a | 57.41 | a | -4.10 | 62.90 | abcdefg | 63.21 | abcd | 0.49 | 62.21 | a | 61.11 | a | -1.77 |
| CIP316361.187 | 61.30 | abc | 62.96 | a | 2.72 | 52.64 | abc | 52.90 | ab | 0.49 | 70.12 | abcd | 64.69 | abcd | -7.75 | 57.09 | ab | 56.85 | ab | -0.41 |
| CIP316360.241 | 33.64 | efghij | 42.16 | abcde | 25.32 | 56.72 | ab | 50.37 | abcd | -11.19 | 38.89 | ghijk | 52.41 | abcd | 34.76 | 60.85 | a | 54.20 | abc | -10.94 |
| CIP316346.204 | 54.07 | abcdef | 55.06 | abcde | 1.83 | 50.44 | abcd | 51.73 | abc | 2.55 | 56.67 | abcdefghi | 61.23 | abcd | 8.06 | 53.47 | abc | 53.21 | abcd | -0.48 |
| CIP316375.102 | 65.86 | a | 67.04 | a | 1.78 | 47.44 | abcde | 48.27 | abcde | 1.74 | 78.09 | ab | 84.88 | a | 8.70 | 50.47 | abcd | 52.10 | abcde | 3.23 |
| CIP316365.166 | 31.11 | fghij | 32.53 | bcde | 4.56 | 49.77 | abcd | 47.04 | abcdef | -5.48 | 35.74 | hijk | 39.32 | cd | 10.02 | 51.86 | abcd | 51.36 | abcde | -0.98 |
| CIP316367.134 | 63.58 | ab | 64.01 | a | 0.68 | 46.63 | abcdef | 39.88 | abcdefgh | -14.48 | 78.46 | a | 77.04 | ab | -1.81 | 55.15 | abc | 50.74 | abcde | -7.99 |
| CIP316361.158 | 32.96 | efghij | 42.22 | abcde | 28.09 | 44.17 | abcdefg | 45.99 | abcdef | 4.11 | 36.42 | hijk | 47.16 | bcd | 29.49 | 46.70 | abcdef | 49.94 | abcde | 6.93 |
| CIP316361.191 | 54.75 | abcdef | 58.70 | abc | 7.22 | 43.51 | abcdefgh | 44.94 | abcdefgh | 3.29 | 61.79 | abcdefg | 73.46 | abc | 18.88 | 47.77 | abcde | 49.01 | abcdef | 2.61 |
| CIP316356.149 | 52.47 | abcdef | 57.10 | abcd | 8.82 | 37.17 | cdefghij | 44.57 | cdefghij | 19.89 | 56.98 | abcdefghi | 62.72 | abcd | 10.08 | 39.15 | cdefgh | 47.72 | abcdefg | 21.89 |
| CIP316361.244 | 59.44 | abcd | 32.78 | bcde | -44.86 | 48.58 | abcde | 44.32 | abcde | -8.77 | 65.06 | abcdefg | 45.62 | bcd | -29.89 | 52.04 | abc | 46.98 | abcdefg | -9.73 |
| CIP316367.147 | 45.52 | abcdefg | 66.67 | a | 46.46 | 42.36 | bcdefghi | 39.94 | abcdefgh | -5.71 | 73.89 | abc | 74.81 | ab | 1.25 | 47.36 | abcde | 43.70 | abcdefgh | -7.72 |
| CIP316355.162 | 40.80 | bcdefgh | 57.84 | abcd | 41.75 | 38.06 | cdefghij | 39.38 | abcdefgh | 3.47 | 48.15 | cdefghi | 61.67 | abcd | 28.08 | 40.22 | bcdefgh | 43.33 | abcdefgh | 7.73 |
| Yungay | 9.01 | j | 45.43 | abcde | 404.11 | 1.02 | n | 28.70 | fghi | 2701.18 | 14.88 | k | 58.70 | abcd | 294.58 | 4.17 | k | 43.15 | abcdefgh | 934.03 |
| CIP316361.190 | 46.05 | abcdefg | 47.65 | abcde | 3.49 | 37.11 | cdefghij | 38.52 | bcdefghi | 3.79 | 67.72 | abcdef | 75.56 | ab | 11.58 | 41.49 | bcdefgh | 42.35 | abcdefgh | 2.05 |
| CIP316352.122 | 18.70 | hij | 50.43 | abcde | 169.64 | 32.33 | efghijk | 37.16 | bcdefghi | 14.93 | 23.58 | jk | 52.90 | abcd | 124.35 | 34.37 | defghi | 41.73 | abcdefgh | 21.41 |
| CIP316361.118 | 52.78 | abcdef | 55.06 | abcde | 4.33 | 37.54 | cdefghij | 38.64 | abcdefghij | 2.93 | 57.53 | abcdefghi | 70.06 | abc | 21.78 | 38.59 | cdefghi | 41.60 | abcdefgh | 7.81 |
| CIP316367.118 | 60.99 | abc | 61.73 | ab | 1.21 | 37.17 | cdefghij | 35.12 | bcdefghi | -5.51 | 68.95 | abcde | 75.80 | ab | 9.94 | 40.51 | bcdefg | 41.05 | abcdefgh | 1.34 |
| CIP316367.177 | 43.27 | abcdefgh | 60.74 | ab | 40.37 | 41.67 | bcdefghi | 36.91 | bcdefghi | -11.41 | 47.41 | defghi | 70.43 | abc | 48.57 | 45.49 | abcdefg | 39.81 | bcdefgh | -12.48 |
| CIP316353.741 | 47.59 | abcdefg | 43.09 | abcde | -9.47 | 37.46 | cdefghij | 35.93 | bcdefghi | -4.09 | 60.86 | abcdefgh | 59.69 | abcd | -1.93 | 40.17 | bcdefgh | 39.57 | bcdefgh | -1.51 |
| CIP316367.148 | 54.69 | abcdef | 61.48 | ab | 12.42 | 35.40 | defghij | 34.88 | bcdefghi | -1.47 | 59.20 | abcdefgh | 69.51 | abc | 17.41 | 40.02 | bcdefgh | 38.83 | bcdefgh | -2.99 |
| CIP316354.112 | 38.21 | cdefghi | 56.85 | abcd | 48.79 | 26.48 | ijkl | 31.85 | defghi | 20.28 | 45.74 | efghij | 65.00 | abcd | 42.11 | 28.89 | ghi | 38.52 | bcdefgh | 33.33 |
| CIP316361.121 | 24.63 | ghij | 28.46 | de | 15.54 | 35.95 | cdefghij | 36.73 | bcdefghi | 2.16 | 29.44 | ijk | 33.89 | d | 15.09 | 38.05 | cdefghi | 38.40 | bcdefgh | 0.91 |
| CIP316361.209 | 20.06 | hij | 29.20 | cde | 45.54 | 35.51 | defghij | 33.33 | cdefghi | -6.12 | 25.62 | ijk | 38.89 | cd | 51.81 | 37.91 | cdefghi | 38.33 | bcdefgh | 1.11 |
| CIP316353.148 | 52.65 | abcdef | 53.77 | abcde | 2.11 | 35.78 | cdefghij | 36.23 | bcdefghi | 1.28 | 59.14 | abcdefgh | 61.73 | abcd | 4.38 | 37.69 | cdefghi | 37.72 | bcdefgh | 0.07 |
| CIP316344.165 | 34.57 | defghi | 25.12 | e | -27.32 | 26.72 | hijkl | 28.83 | fghi | 7.90 | 43.40 | fghij | 32.90 | d | -24.18 | 31.47 | efghij | 35.19 | cdefgh | 11.81 |
| Amarilis | 14.14 | ij | 48.46 | abcde | 242.79 | 6.09 | mn | 28.70 | fghi | 371.60 | 23.77 | jk | 55.49 | abcd | 133.51 | 10.72 | jk | 34.44 | cdefgh | 221.43 |

*(Continued)*

**Table 5.** (Continued)

| Clone | Marketable tuber yield t/ha | | | | | | | | | | Total tuber yield t/ha | | | | | | | | | |
|---|---|---|---|---|---|---|---|---|---|---|---|---|---|---|---|---|---|---|---|---|
| | Huanuco | | | | | Oxapampa | | | | | Huanuco | | | | | Oxapampa | | | | |
| | Without control | α = 0.05 | With control | α = 0.05 | Increase % | Without control | α = 0.05 | With control | α = 0.05 | Increase % | Without control | α = 0.05 | With control | α = 0.05 | Increase % | Without control | α = 0.05 | With control | α = 0.05 | Increase % |
| CIP316354.169 | 49.81 | abcdef | 53.95 | abcde | 8.30 | 28.20 | ghijkl | 29.69 | efghi | 5.30 | 54.38 | bcdefghi | 60.06 | abcd | 10.44 | 29.68 | fghij | 33.52 | defgh | 12.94 |
| Kory | 37.65 | cdefghi | 41.85 | abcde | 11.15 | 17.05 | klm | 25.43 | hi | 49.17 | 46.73 | defghi | 64.94 | abcd | 38.97 | 21.21 | hijk | 32.22 | efgh | 51.92 |
| CIP316358.214 | 50.00 | abcdef | 52.72 | abcde | 5.43 | 26.72 | hijkl | 27.10 | ghi | 1.43 | 54.57 | bcdefghi | 57.78 | abcd | 5.88 | 29.25 | fghij | 29.51 | fgh | 0.89 |
| CIP316352.152 | 37.84 | cdefghi | 40.86 | abcde | 7.99 | 21.93 | jklm | 22.65 | hi | 3.32 | 43.70 | fghijk | 52.28 | abcd | 19.63 | 24.27 | hij | 29.44 | fgh | 21.31 |
| CIP316375.101 | 48.09 | abcdefg | 50.56 | abcde | 5.13 | 29.79 | fghijk | 26.11 | ghi | −12.35 | 62.53 | abcdefg | 66.36 | abcd | 6.12 | 32.32 | efghij | 28.52 | gh | −11.76 |
| CIP316387.156 | 37.72 | cdefghi | 48.89 | abcde | 29.62 | 13.98 | lm | 20.06 | i | 43.55 | 40.93 | fghijk | 52.65 | abcd | 28.66 | 14.96 | ijk | 23.52 | h | 57.18 |
| Average clones | 43.59 | | 50.44 | | 36.80 | 37.18 | | 39.02 | | 104.64 | 51.53 | | 60.38 | | 30.49 | 40.47 | | 43.54 | | 42.88 |
| Average varieties | 41.21 | | 46.77 | | 14.25 | 21.90 | | 22.94 | | 11.51 | 49.05 | | 57.10 | | 18.14 | 23.85 | | 27.16 | | 22.24 |
| Standard deviation | 4.47 | | 5.04 | | 2.80 | 2.80 | | 3.40 | | | 3.99 | | 5.55 | | | 2.93 | | 3.65 | | |

Different letter after the mean value indicates a significant difference according to Tukey test (P < 0.05).

**Table 6. Environmental Impact Rate in clones and varieties control in Huánuco and Oxapampa 2021–2022.**

| Clone | Number of Applications | | Environmental Impact Rate (EIR) | | | | | | | | Aver-age | Reduction of EIR (%) |
| --- | --- | --- | --- | --- | --- | --- | --- | --- | --- | --- | --- | --- |
| | | | Huánuco | | | | Oxapampa | | | | | |
| | Huánuco | Oxapampa | Man-cozeb | Cymox-anil | Propineb | Total | Man-cozeb | Cymox-anil | Propineb | Total | | |
| CIP316367.147 | Mancozeb = 2 | Mancozeb = 2 | 46.72 | 0.00 | 0.00 | 46.72 | 46.72 | 0.00 | 0.00 | 46.72 | 46.72 | 0.82 |
| CIP316352.152 | Mancozeb = 2 | Mancozeb = 3 | 46.72 | 0.00 | 0.00 | 46.72 | 70.08 | 0.00 | 0.00 | 70.08 | 58.40 | 0.77 |
| CIP316353.741 | Mancozeb = 2 | Mancozeb = 3 | 46.72 | 0.00 | 0.00 | 46.72 | 70.08 | 0.00 | 0.00 | 70.08 | 58.40 | 0.77 |
| CIP316354.169 | Mancozeb = 2 | Mancozeb = 3 | 46.72 | 0.00 | 0.00 | 46.72 | 70.08 | 0.00 | 0.00 | 70.08 | 58.40 | 0.77 |
| CIP316356.149 | Mancozeb = 2 | Mancozeb = 3 | 46.72 | 0.00 | 0.00 | 46.72 | 70.08 | 0.00 | 0.00 | 70.08 | 58.40 | 0.77 |
| CIP316358.214 | Mancozeb = 2 | Mancozeb = 3 | 46.72 | 0.00 | 0.00 | 46.72 | 70.08 | 0.00 | 0.00 | 70.08 | 58.40 | 0.77 |
| CIP316360.241 | Mancozeb = 2 | Mancozeb = 3 | 46.72 | 0.00 | 0.00 | 46.72 | 70.08 | 0.00 | 0.00 | 70.08 | 58.40 | 0.77 |
| CIP316361.118 | Mancozeb = 2 | Mancozeb = 3 | 46.72 | 0.00 | 0.00 | 46.72 | 70.08 | 0.00 | 0.00 | 70.08 | 58.40 | 0.77 |
| CIP316361.121 | Mancozeb = 2 | Mancozeb = 3 | 46.72 | 0.00 | 0.00 | 46.72 | 70.08 | 0.00 | 0.00 | 70.08 | 58.40 | 0.77 |
| CIP316361.187 | Mancozeb = 2 | Mancozeb = 3 | 46.72 | 0.00 | 0.00 | 46.72 | 70.08 | 0.00 | 0.00 | 70.08 | 58.40 | 0.77 |
| CIP316361.190 | Mancozeb = 2 | Mancozeb = 3 | 46.72 | 0.00 | 0.00 | 46.72 | 70.08 | 0.00 | 0.00 | 70.08 | 58.40 | 0.77 |
| CIP316361.191 | Mancozeb = 2 | Mancozeb = 3 | 46.72 | 0.00 | 0.00 | 46.72 | 70.08 | 0.00 | 0.00 | 70.08 | 58.40 | 0.77 |
| CIP316361.244 | Mancozeb = 2 | Mancozeb = 3 | 46.72 | 0.00 | 0.00 | 46.72 | 70.08 | 0.00 | 0.00 | 70.08 | 58.40 | 0.77 |
| CIP316367.118 | Mancozeb = 2 | Mancozeb = 3 | 46.72 | 0.00 | 0.00 | 46.72 | 70.08 | 0.00 | 0.00 | 70.08 | 58.40 | 0.77 |
| CIP316367.134 | Mancozeb = 2 | Mancozeb = 3 | 46.72 | 0.00 | 0.00 | 46.72 | 70.08 | 0.00 | 0.00 | 70.08 | 58.40 | 0.77 |
| CIP316367.148 | Mancozeb = 2 | Mancozeb = 3 | 46.72 | 0.00 | 0.00 | 46.72 | 70.08 | 0.00 | 0.00 | 70.08 | 58.40 | 0.77 |
| CIP316375.101 | Mancozeb = 2 | Mancozeb = 3 | 46.72 | 0.00 | 0.00 | 46.72 | 70.08 | 0.00 | 0.00 | 70.08 | 58.40 | 0.77 |
| CIP316344.165 | Mancozeb = 2 | Mancozeb = 4 | 46.72 | 0.00 | 0.00 | 46.72 | 93.44 | 0.00 | 0.00 | 93.44 | 70.08 | 0.72 |
| CIP316346.204 | Mancozeb = 2 | Mancozeb = 4 | 46.72 | 0.00 | 0.00 | 46.72 | 93.44 | 0.00 | 0.00 | 93.44 | 70.08 | 0.72 |
| CIP316352.122 | Mancozeb = 3 | Mancozeb = 3 | 70.08 | 0.00 | 0.00 | 70.08 | 70.08 | 0.00 | 0.00 | 70.08 | 70.08 | 0.72 |
| CIP316353.148 | Mancozeb = 2 | Mancozeb = 4 | 46.72 | 0.00 | 0.00 | 46.72 | 93.44 | 0.00 | 0.00 | 93.44 | 70.08 | 0.72 |
| CIP316354.112 | Mancozeb = 2 | Mancozeb = 4 | 46.72 | 0.00 | 0.00 | 46.72 | 93.44 | 0.00 | 0.00 | 93.44 | 70.08 | 0.72 |
| CIP316355.162 | Mancozeb = 2 | Mancozeb = 4 | 46.72 | 0.00 | 0.00 | 46.72 | 93.44 | 0.00 | 0.00 | 93.44 | 70.08 | 0.72 |
| CIP316361.209 | Mancozeb = 3 | Mancozeb = 3 | 70.08 | 0.00 | 0.00 | 70.08 | 70.08 | 0.00 | 0.00 | 70.08 | 70.08 | 0.72 |
| CIP316365.166 | Mancozeb = 3 | Mancozeb = 3 | 70.08 | 0.00 | 0.00 | 70.08 | 70.08 | 0.00 | 0.00 | 70.08 | 70.08 | 0.72 |
| CIP316367.117 | Mancozeb = 2 | Mancozeb = 4 | 46.72 | 0.00 | 0.00 | 46.72 | 93.44 | 0.00 | 0.00 | 93.44 | 70.08 | 0.72 |
| CIP316375.102 | Mancozeb = 2 | Mancozeb = 4 | 46.72 | 0.00 | 0.00 | 46.72 | 93.44 | 0.00 | 0.00 | 93.44 | 70.08 | 0.72 |
| CIP316387.156 | Mancozeb = 2 | Mancozeb = 4 | 46.72 | 0.00 | 0.00 | 46.72 | 93.44 | 0.00 | 0.00 | 93.44 | 70.08 | 0.72 |
| Kory | Mancozeb = 2 | Mancozeb = 2 Cymox-anil = 2 Propineb = 2 | 46.72 | 0.00 | 0.00 | 46.72 | 46.72 | 2.09 | 47.32 | 96.13 | 71.42 | 0.72 |
| CIP316361.158 | Mancozeb = 3 | Mancozeb = 4 | 70.08 | 0.00 | 0.00 | 70.08 | 93.44 | 0.00 | 0.00 | 93.44 | 81.76 | 0.68 |
| CIP316367.177 | Mancozeb = 3 | Mancozeb = 4 | 70.08 | 0.00 | 0.00 | 70.08 | 93.44 | 0.00 | 0.00 | 93.44 | 81.76 | 0.68 |
| Amarilis | Mancozeb = 4 Cymox-anil = 4 Propineb = 4 | Mancozeb = 3 Cymox-anil = 5 Propineb = 5 | 46.72 | 4.18 | 94.64 | 145.54 | 70.08 | 5.22 | 118.30 | 193.60 | 169.57 | 0.33 |
| Yungay | Mancozeb = 5 Cymox-anil = 7 Propineb = 7 | Mancozeb = 3 Cymox-anil = 6 Propineb = 6 | 116.80 | 7.31 | 165.62 | 289.73 | 70.08 | 6.26 | 141.96 | 218.30 | 254.02 | 0.00 |

EIC: Mancozeb = 14.60, Cymoxanil = 8.7, Propineb = 16.9.

Active Ingredient: Mancozeb = 80%, Cymoxanil = 0.06%, Propineb = 70%.

Dose: 2 Kg/ha.

Yungay. This last parameter must have a lower value than that of the Yungay, which had an MYSI of 31 (Table 9).

Fourteen clones had the lowest MYSI that ranged from 5 to 29 and whose marketable tuber yield under late blight control ranged from 40.6 to 63.7 t/ha. According to the AMMI analysis,

**Table 7. Economic profitability of potato clones resistant to late blight (2021–2022).**

| Variety | Late blight resistant breeding clones | Kory (R) | Amarilis (MR) | Yungay (S) |
|---|---|---|---|---|
| Economic profitability | 189.80 | 70.46 | 45.45 | 36.50 |

R = Resistant to Late blight, MR = Moderately resistant, S= Susceptible.

**Table 8. Additive main effects multiplicative interaction (AMMI) model analysis for marketable tuber yield (2021–2022).**

| Source of variation | Degrees of freedom | Mean square | Contribution | |
|---|---|---|---|---|
| | | | % | Accumulated (%) |
| Environments | 2 | 4374.6o** | | |
| Blocks/Environments | 6 | 171.20 | | |
| Clones | 32 | 488.80** | | |
| Clones × Environments | 64 | 272.10** | | |
| CP1 | 33 | 326.50** | 61.90 | 61.90 |
| CP2 | 31 | 214.10** | 38.10 | 100.00 |
| Error | 192 | 70.30 | | |
| Total | 296 | | | |
| Coefficient of variation (%) | 18.59 | | | |

**indicates significant source of variation at $P \leq 0.01$.

these clones had a lower GE interaction or at least lower than the GE interaction of the susceptible Yungay (Table 9, Fig 3). The breeding clone CIP316375.102 had the highest yield of marketable tubers (63.7 t/ha) higher than the yields of Yungay, Kory and Amarilis by 41.5, 40.6 and 40.9 t/ha, respectively. The clones with phenotypic stability are those that have had the most stable marketable tuber yields throughout the three locations. Fig 3 shows the biplot of the marketable yield against the main component 1 (CP1), which contributed with 61.9% of the GE interaction. The breeding clone CIP316361.209 is not phenotypically stable, but is better adapted to El Mantaro (Fig 3).

## Discussion

The AUDPC and sAUDPC were lower in Huánuco probably because of the weather conditions of temperature, relative humidity and precipitation, as well as the probable presence of *P. infestans* isolates different from those in Oxapampa according to [17]. However, these weather conditions were sufficient to induce a high disease pressure in the susceptible variety Yungay. However, it was necessary to control late blight in a timely and adequate manner to achieve economically profitable yields. Twelve applications were carried out during the vegetative period in Huánuco and nine in Oxapampa coinciding with [10], who mentions that in some places with high disease pressure up to 20 applications are needed during the vegetative period of the crop, thus causing a high rate of environmental impact due to the greater use of fungicides compared to clones that received two to four applications depending on their level of resistance as mentioned by [20,21]. In Huánuco, the environmental impact rate of resistant clones to late blight was 6 times less than the EIR of Yungay and three times less than Amarilis, while in Oxapampa it was 4 and 2.5 times with respect to the EIR of Yungay and Amarilis respectively, because of the lower use of fungicides.

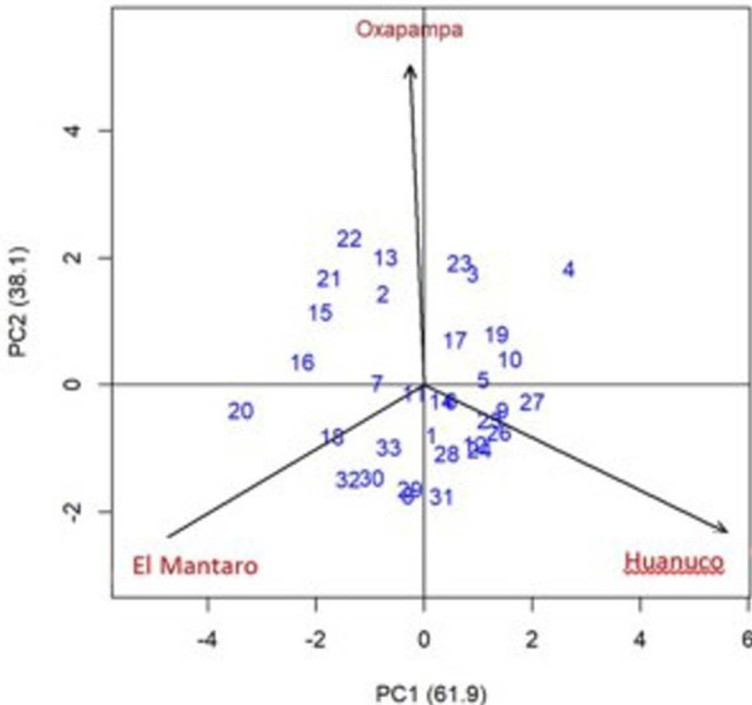

**Fig 2. Biplot using principal components 1 and 2 (CP1, CP2) of 30 clones, three control varieties and three locations for marketable tuber yield (2021–2022).**

The low EIR found in the clones resistant to late blight, due to the lower use of fungicides to control the disease, compared to the susceptible varieties Yungay and Amarilis that had a high IER, due to the greater amount of application of fungicides used for the control of late blight and that allows obtaining economically profitable yields, while the clones with a minimum use of fungicides obtained high tuber yield, even if we compare the yield of the clones in experiment 1, where the clones and varieties only received two applications of Mancozeb and the yield in experiment 2, where the clones received two to four applications of Mancozeb and the varieties up to twelve applications of contact and systemic fungicides, the increase in the yield of the clones was very low compared to the varieties that did have a significant increase in their yields, at the cost of a high EIR, coinciding with [19], who mentions that the high use of agrochemicals puts public health at risk and its impact on the environment,.

The clone CIP396367.147 managed to reduce the EIR by 82% with respect to the EIR of Yungay, while the rest of the clones reduced the EIR between 68 to 77%. This information allows us to verify that it is possible to reduce the environmental impact rate with the use of late blight-resistant potato clones that require a minimum of applications with contact fungicides to achieve economically profitable yields, for the benefit of preserving the environment. The health of producers also benefits by being less exposed to contact with agrochemicals and consumers by having a product with minimal agrochemical residues. In addition, the lower use of fungicides in resistant clones allows to reduce production costs and therefore increase profitability. These clones can be incorporated into sustainable production systems such as family farming since they have positive effects on the environment, increase profitability and improve the quality of life of producers.

**Table 9. Principal components AMMI, Marketable Yield (MY), Stability value AMMI (SVAMMI), Marketable Yield stability Index (MYSI), AUDPC, sAUDPC, Environmental Impact Rate (EIR) in Huánuco, Oxapampa and El Mantaro (Junin) (2021–2022).**

| # | Clone | PC1 | PC2 | MY t/ha. | SVAMMI | MYSI | AUDPC | sAUDPC | EIR | Reduction of EIR (%) | Phenotypic Stability |
|---|---|---|---|---|---|---|---|---|---|---|---|
| 1 | CIP316375.102 | −0.97 | −1.45 | 63.70 | 2.14 | 18 | 204 | 0.72 | 70.08 | 0.72 | Si |
| 2 | CIP316361.187 | 0.57 | 0.72 | 56.60 | 1.17 | 7 | 206 | 0.73 | 58.40 | 0.77 | Si |
| 3 | CIP316367.117 | 0.66 | 1.92 | 54.40 | 2.20 | 21 | 143 | 0.51 | 70.08 | 0.72 | Si |
| 4 | CIP316356.149 | −0.18 | −0.13 | 52.80 | 0.32 | 5 | 262 | 0.93 | 58.40 | 0.77 | Si |
| 5 | CIP316367.147 | 1.39 | −0.74 | 52.00 | 2.37 | 27 | 163 | 0.63 | 46.72 | 0.82 | Si |
| 6 | CIP316367.134 | 1.21 | −0.55 | 50.80 | 2.04 | 23 | 99 | 0.52 | 58.40 | 0.77 | Si |
| 7 | CIP316367.177 | 0.43 | −1.07 | 50.60 | 1.28 | 14 | 372 | 1.32 | 81.76 | 0.68 | Si |
| 8 | CIP316346.204 | 0.89 | 1.76 | 49.60 | 2.28 | 29 | 111 | 0.39 | 70.08 | 0.72 | Si |
| 9 | CIP316354.112 | −0.30 | −1.74 | 49.00 | 1.81 | 23 | 219 | 0.78 | 70.08 | 0.72 | Si |
| 10 | CIP316367.132 | 1.02 | −1.01 | 48.40 | 1.94 | 27 | 178 | 0.35 | 58.40 | 0.77 | Si |
| 11 | CIP316361.118 | 0.32 | −0.25 | 47.60 | 0.57 | 16 | 162 | 0.57 | 58.40 | 0.77 | Si |
| 12 | CIP316353.148 | 0.51 | −0.25 | 45.30 | 0.86 | 20 | 207 | 0.73 | 70.08 | 0.72 | Si |
| 13 | CIP316353.741 | −0.87 | 0.03 | 43.00 | 1.41 | 26 | 149 | 0.53 | 58.40 | 0.77 | Si |
| 14 | CIP316375.101 | −0.26 | −1.64 | 42.80 | 1.69 | 29 | 94 | 0.33 | 58.40 | 0.77 | Si |
| 15 | Amarilis | −0.14 | 0.78 | 40.60 | 0.81 | 29 | 1126 | 3.99 | 169.57 | 0.33 | Si |
| 16 | Yungay | −0.65 | −0.98 | 41.50 | 1.44 | 31 | 1441 | 6.00 | 254.02 | 0.00 | No |
| 17 | CIP316361.191 | 1.33 | 0.81 | 48.30 | 2.30 | 33 | 88 | 0.31 | 58.40 | 0.77 | No |
| 18 | CIP316361.190 | −1.71 | −0.81 | 50.10 | 2.89 | 35 | 207 | 0.74 | 58.40 | 0.77 | No |
| 19 | CIP316360.241 | −0.72 | 2.02 | 46.40 | 2.33 | 36 | 126 | 0.45 | 58.40 | 0.77 | No |
| 20 | CIP316361.158 | −2.27 | 0.37 | 50.90 | 3.70 | 37 | 467 | 1.66 | 81.76 | 0.68 | No |
| 21 | CIP316358.214 | 0.93 | −0.93 | 40.10 | 1.77 | 38 | 239 | 0.85 | 58.40 | 0.77 | No |
| 22 | CIP316355.162 | 1.58 | 0.41 | 45.00 | 2.60 | 41 | 226 | 0.80 | 70.08 | 0.72 | No |
| 23 | CIP316352.152 | 1.09 | 0.09 | 30.00 | 1.77 | 42 | 189 | 0.67 | 58.40 | 0.77 | No |
| 24 | CIP316387.156 | 0.33 | −1.75 | 37.50 | 1.83 | 42 | 318 | 1.13 | 70.08 | 0.72 | No |
| 25 | CIP316367.148 | 2.00 | −0.26 | 44.60 | 3.26 | 46 | 161 | 0.63 | 58.40 | 0.77 | No |
| 26 | CIP316344.165 | −0.76 | 1.44 | 28.10 | 1.89 | 47 | 225 | 0.80 | 70.08 | 0.72 | No |
| 27 | Kory | −1.40 | −1.48 | 40.90 | 2.71 | 50 | 353 | 1.25 | 71.42 | 0.72 | No |
| 28 | CIP316354.169 | 1.45 | −0.39 | 39.80 | 2.39 | 51 | 213 | 0.76 | 58.40 | 0.77 | No |
| 29 | CIP316361.244 | −1.75 | 1.69 | 41.90 | 3.31 | 51 | 64 | 0.23 | 58.40 | 0.77 | No |
| 30 | CIP316365.166 | −1.38 | 2.32 | 41.30 | 3.22 | 51 | 420 | 1.49 | 70.08 | 0.72 | No |
| 31 | CIP316361.209 | −3.40 | −0.40 | 42.20 | 5.53 | 54 | 428 | 1.52 | 70.08 | 0.72 | No |
| 32 | CIP316361.121 | −1.93 | 1.16 | 37.30 | 3.34 | 60 | 383 | 1.36 | 58.40 | 0.77 | No |
| 33 | CIP316352.122 | 2.68 | 1.84 | 35.10 | 4.72 | 63 | 520 | 1.84 | 70.08 | 0.72 | No |

The profitability of late blight resistant clones was higher than the control varieties, due to the lower use of fungicides, labor and the higher tuber yield compared to the lower yield in the susceptible control varieties such as Yungay and Amarilis, allowing producers to significantly increase their profitability when using these clones in commercial production.

The stability of tuber yield in clones with resistance to late blight is very important. In the AMMI analysis, clones with MTSI values lower and higher than the susceptible variety Yungay were found, one of the varieties widely planted by farmers and preferred by consumers in Peru. According to [42], clones with low MTSI values are those that are more stable and with high yields, considering this criterion for the selection of clones with values below the MTSI value of the Yungay variety.

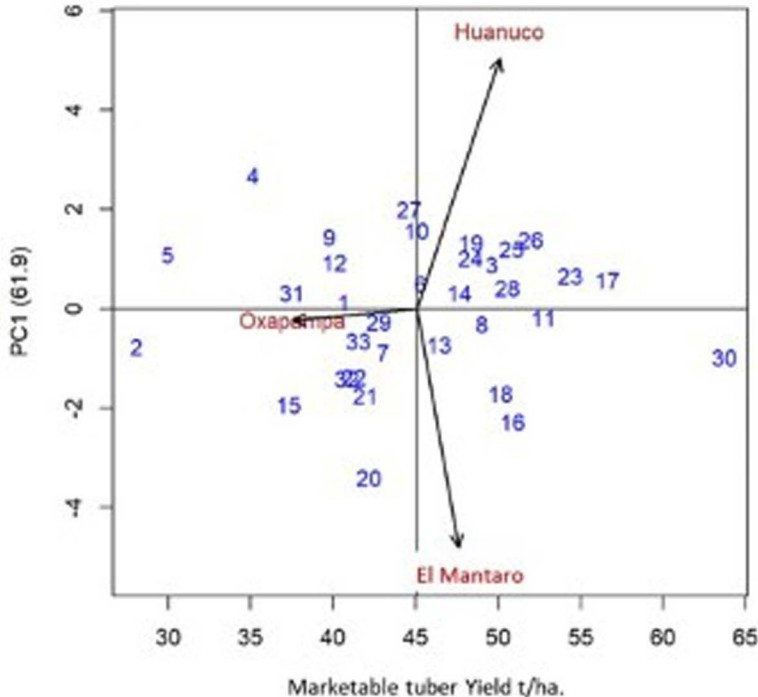

**Fig 3. Biplot of potato clones of marketable tuber yield versus principal component 1 (PC1) in three locations in Peru (2021–2022).**

## Conclusions

Fourteen clones were selected for based on their high resistance to late blight, low EIR, high economic profitability and phenotypic stability for the marketable tuber yield.

Clones CIP316375.102, CIP316361.187, CIP316367.117, CIP316356.149, CIP316367.147 were the ones that presented the highest tuber yield, phenotypically stable, high resistance to Late blight, low environmental impact and high economic profitability, superior to control cultivars.

These potential new cultivars could contribute to preserving the environment, while also being economically profitable, this would improve the standard of living, particularly for small and medium-sized potato producers in Peru.

## Acknowledgments

To Engineers Alejandro Mendoza and Giovana Panduro who supported the field trials in Huánuco and Oxapampa respectively, and to Dr. Moctar Kante for his support in reviewing the manuscript.

## Author contributions

**Conceptualization:** Manuel Gastelo.

**Investigation:** Manuel Gastelo, Carolina Bastos.

**Supervision:** Manuel Gastelo, Raul Blas.

**Writing – original draft:** Manuel Gastelo.

**Writing – review & editing:** Manuel Gastelo, Rodomiro Ortiz, Raul Blas.

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
