## [Decision Letter · Decision Letter 0]

13 Nov 2024

PONE-D-24-45529Environmental impact and phenotypic stability in potato clones resistant to late blight Phytophthora infestans (Mont) de Bary, resilient to climate change in PeruPLOS ONE

Dear Dr. Gastelo,

Thank you for submitting your manuscript to PLOS ONE. After careful consideration, we feel that it has merit but does not fully meet PLOS ONE’s publication criteria as it currently stands. Therefore, we invite you to submit a revised version of the manuscript that addresses the points raised during the review process.

We look forward to receiving your revised manuscript.

Kind regards,

Ravinder Kumar, Ph.D.

Academic Editor

PLOS ONE

“USAID”

Reviewers' comments:

Reviewer's Responses to Questions

**Comments to the Author**

1. Is the manuscript technically sound, and do the data support the conclusions?

Reviewer #1: Yes

Reviewer #2: Yes

2. Has the statistical analysis been performed appropriately and rigorously? 

Reviewer #1: Yes

Reviewer #2: Yes

3. Have the authors made all data underlying the findings in their manuscript fully available?

Reviewer #1: Yes

Reviewer #2: Yes

4. Is the manuscript presented in an intelligible fashion and written in standard English?

Reviewer #1: Yes

Reviewer #2: Yes

5. Review Comments to the Author

Reviewer #1: Dear Authors,

After careful reviewing the manuscript `Environmental impact and phenotypic stability in potato clones resistant to late blight Phytophthora infestans (Mont) de Bary, resilient to climate change in Peru`, I congratulate you and kindly consider the following recommendations:

- The name of Figure 1 must be translated in English.

- I propose rearranging the information in Table 3. Everywhere, the same values are repeated twice. I advise you to review the results just once and note that they apply to both scenarios with and without the control.

- Line 87: the source Villazón et al. 2005 isn't listed on References list. Nevertheless there is a Villamon et al. Is it the same author? Please update the list.

- Line 135: the statistical `package` is actually the software you used. You didn't mentioned any package for them, so please correct the line, replace package with software.

Reviewer #2: Reviewer Comments to Author:

Ms. Ref. No.: PONE-D-24-45529

Ms. Full Title: Environmental impact and phenotypic stability in potato clones resistant to late blight Phytophthora infestans (Mont) de Bary, resilient to climate change in Peru

The authors conducted a study on the Environmental impact and phenotypic stability in potato clones resistant to late blight. Using total tuber yield, marketable tuber yield, evolution of disease intensity over time (AUDPC, sAUDPC), they analyzed a comprehensive 30 potatoes clones for phenotypic stability and economic profitability across locations. Late blight resistance was evaluated in a trial with no chemical application while the application of chemical in controlling late blight helped in evaluation environmental impact. The study was carried out on one season across several location, which is an admirable effort. However, substantial revisions are required if the manuscript is to be considered for publication in this valued journal.

Abstract:

1. Line 19. You mentioned that the cultivars Yungay, Amaryllis and Kory were used as controls. Are they controls in terms of resistance to late blight, in terms of yield stability or both? It is important to clarify

2. Line 21. You mention that fourteen clones were resistant, stable phenotypically, low environmental impact and high profitability. Are they having all those characteristics at the same level? Likely not. It is good to mention the main clones. You can mention the top five if you can’t mention all in decreasing levels

Introduction:

1. In the last paragraph in the introduction part containing the aims of the study. The authors should change the objectives and project there the different hypothesis that are tested in the study emphatically.

2. The novelty of the study over published literature has to be highlighted in the last paragraph of introduction.

Materials and methods:

1. Line 76. You mentioned that the 30 genotypes tested are resistant to late blight. At 850 masl altitude. What was the basis of the selection of those genotypes? Why are you screening against for the resistance? If there is any reason for that, it has to be clearly hypothesized at the introduction

2. Line 79: The controls you used: Amarilis (moderately resistant), Yungay (susceptible) and Kory (resistant). These characteristics where supposed to be the same when using those varieties at your study site?

3. Line 86: precise the precipitation, relative humidity and temperature that give these optimal conditions

4. Line 141-142. You wrote environmental Impact rate (IA) . IA is it the abbreviation?

5. Line 143: All the elements for the EIR formula are not defined. Defined PAI and NA on the formula

Results:

1. Line 220, Table 4. Values for AUDPC and sAUDPC are they fixed values or means of values? If they are means of values, provides Standard Deviation so that we can appreciate the significance from the Tukey means separation test. If is fixed values, how do the Tukey test was performed?

2. Line 254: The title of Figure 1 is in Spanish, not in English. You have to harmonize

3. Line 271. Table 5. Why you dis not performed the Tukey Test for the mean separation in order to compare clones as you did with results on AUDPC and sAUDPC provided in Table 4?

4. Table 5: Can we also identify resistant clones from this Table? Are they those with less increase when controlling the disease? If so, provide a relationship between AUDPC values and increase in yield

5. Line 274. Results presented in Figure 2 look appearing in Table 5. Duplication of results. You should choose to presents those results in a Table or in Figure

6. Line 312. Table 6. Could you provide mean separation for EIR for the different genotypes? It will help differentiating the different clones

Conclusion:

Provide a conclusion for your research. You should come out with the most promising clones resistant to late blight, with important yield, stable across environments and with higher economic profitability

6. PLOS authors have the option to publish the peer review history of their article (what does this mean? ). If published, this will include your full peer review and any attached files.

**Do you want your identity to be public for this peer review?** For information about this choice, including consent withdrawal, please see our Privacy Policy .

Reviewer #1: No

Reviewer #2: **Yes: ** Eric Bertrand Kouam

---

## [Author Response · Author response to Decision Letter 0]

11 Dec 2024

Response to Academic Editor and Reviewers

DONE

“USAID”

DONE

DONE

Reviewer #1:

After careful reviewing the manuscript `Environmental impact and phenotypic stability in potato clones resistant to late blight Phytophthora infestans (Mont) de Bary, resilient to climate change in Peru`, I congratulate you and kindly consider the following recommendations:

- The name of Figure 1 must be translated in English.

DONE

- I propose rearranging the information in Table 3. Everywhere, the same values are repeated twice. I advise you to review the results just once and note that they apply to both scenarios with and without the control.

DONE

- Line 87: the source Villazón et al. 2005 isn't listed on References list. Nevertheless there is a Villamon et al. Is it the same author? Please update the list.

DONE

- Line 135: the statistical `package` is actually the software you used. You didn't mentioned any package for them, so please correct the line, replace package with software.

DONE

Reviewer #2:

Ms. Full Title: Environmental impact and phenotypic stability in potato clones resistant to late blight Phytophthora infestans (Mont) de Bary, resilient to climate change in Peru

The authors conducted a study on the Environmental impact and phenotypic stability in potato clones resistant to late blight. Using total tuber yield, marketable tuber yield, evolution of disease intensity over time (AUDPC, sAUDPC), they analyzed a comprehensive 30 potatoes clones for phenotypic stability and economic profitability across locations. Late blight resistance was evaluated in a trial with no chemical application while the application of chemical in controlling late blight helped in evaluation environmental impact. The study was carried out on one season across several location, which is an admirable effort. However, substantial revisions are required if the manuscript is to be considered for publication in this valued journal.

Abstract:

1. Line 19. You mentioned that the cultivars Yungay, Amaryllis and Kory were used as controls. Are they controls in terms of resistance to late blight, in terms of yield stability or both? It is important to clarify

This varieties were used as controls for late blight resistance and tuber yield

2. Line 21. You mention that fourteen clones were resistant, stable phenotypically, low environmental impact and high profitability. Are they having all those characteristics at the same level? Likely not. It is good to mention the main clones. You can mention the top five if you can’t mention all in decreasing levels

Clones CIP316375.102, CIP316361.187, CIP316367.117, CIP316356.149, CIP316367.147 were the ones that presented the highest yields

Introduction:

1. In the last paragraph in the introduction part containing the aims of the study. The authors should change the objectives and project there the different hypothesis that are tested in the study emphatically.

The objective was changed according to the hypotheses of the study

2. The novelty of the study over published literature has to be highlighted in the last paragraph of introduction.

DONE

Materials and methods:

1. Line 76. You mentioned that the 30 genotypes tested are resistant to late blight. At 850 masl altitude. What was the basis of the selection of those genotypes? Why are you screening against for the resistance? If there is any reason for that, it has to be clearly hypothesized at the introduction

The importance of controlling late blight in potato crops has been mentioned in the introduction, one way of controlling it is to use resistant varieties that will allow for increased yields, reduce environmental impact due to less use of fungicides, preserve human health and improve economic profitability for the benefit of farmers. At 850 masl, it was corrected to 1850 masl.

2. Line 79: The controls you used: Amarilis (moderately resistant), Yungay (susceptible) and Kory (resistant). These characteristics where supposed to be the same when using those varieties at your study site?

Yes, these are the resistance levels under the conditions of the study sites and were determined in previous trials

3. Line 86: precise the precipitation, relative humidity and temperature that give these optimal conditions

DONE

4. Line 141-142. You wrote environmental Impact rate (IA) . IA is it the abbreviation?

DONE

5. Line 143: All the elements for the EIR formula are not defined. Defined PAI and NA on the formula

Yes, they are defined in the formula: number of applications (NA) and percentage of active ingredient (PAI)

Results:

1. Line 220, Table 4. Values for AUDPC and sAUDPC are they fixed values or means of values? If they are means of values, provides Standard Deviation so that we can appreciate the significance from the Tukey means separation test. If is fixed values, how do the Tukey test was performed?

AUDPC and sAUDPC values are averages of three replicates. The standard deviation was added

2. Line 254: The title of Figure 1 is in Spanish, not in English. You have to harmonize

Changed from Spanish to English

3. Line 271. Table 5. Why you dis not performed the Tukey Test for the mean separation in order to compare clones as you did with results on AUDPC and sAUDPC provided in Table 4?

Tukey mean comparison test (0.05) and standard deviations were added.

4. Table 5: Can we also identify resistant clones from this Table? Are they those with less increase when controlling the disease? If so, provide a relationship between AUDPC values and increase in yield

Yes, clones with a lower increase in tuber yield when the disease is controlled is due to their resistance.

A correlation was made between the AUDPC values and the increases % in tuber yield. There is a high correlation between AUDPC values and marketable tuber yield increases (r = 0.735, Pearson correlation p<=0.01)

5. Line 274. Results presented in Figure 2 look appearing in Table 5. Duplication of results. You should choose to presents those results in a Table or in Figure

Figure 2 was removed

6. Line 312. Table 6. Could you provide mean separation for EIR for the different genotypes? It will help differentiating the different clones

The average shown in Table 6 corresponds to each clone and is the average of the EIR of Oxapampa and Huánuco.

Conclusion:

Provide a conclusion for your research. You should come out with the most promising clones resistant to late blight, with important yield, stable across environments and with higher economic profitability

Conclusions of the study were added

---

## [Decision Letter · Decision Letter 1]

14 Jan 2025

Environmental impact and phenotypic stability in potato clones resistant to late blight Phytophthora infestans (Mont) de Bary, resilient to climate change in Peru

PONE-D-24-45529R1

Dear Dr. Gastelo,

We’re pleased to inform you that your manuscript has been judged scientifically suitable for publication and will be formally accepted for publication once it meets all outstanding technical requirements.

Kind regards,

Ravinder Kumar, Ph.D.

Academic Editor

PLOS ONE

Reviewers' comments:

Reviewer's Responses to Questions

**Comments to the Author**

1. If the authors have adequately addressed your comments raised in a previous round of review and you feel that this manuscript is now acceptable for publication, you may indicate that here to bypass the “Comments to the Author” section, enter your conflict of interest statement in the “Confidential to Editor” section, and submit your "Accept" recommendation.

Reviewer #1: All comments have been addressed

2. Is the manuscript technically sound, and do the data support the conclusions?

Reviewer #1: Yes

3. Has the statistical analysis been performed appropriately and rigorously? 

Reviewer #1: Yes

4. Have the authors made all data underlying the findings in their manuscript fully available?

Reviewer #1: Yes

5. Is the manuscript presented in an intelligible fashion and written in standard English?

Reviewer #1: Yes

6. Review Comments to the Author

Reviewer #1: (No Response)

7. PLOS authors have the option to publish the peer review history of their article (what does this mean? ). If published, this will include your full peer review and any attached files.

**Do you want your identity to be public for this peer review?** For information about this choice, including consent withdrawal, please see our Privacy Policy .

Reviewer #1: No

---

## [Editor Report · Acceptance letter]

PONE-D-24-45529R1

PLOS ONE

Dear Dr. Gastelo,

I'm pleased to inform you that your manuscript has been deemed suitable for publication in PLOS ONE. Congratulations! Your manuscript is now being handed over to our production team.

Kind regards,

on behalf of

Dr. Ravinder Kumar

Academic Editor

PLOS ONE